# Diagnosing Autism Spectrum Disorders in Deaf Children Using Two Standardised Assessment Instruments: The ADIR-Deaf Adaptation and the ADOS-2 Deaf Adaptation

**DOI:** 10.3390/jcm10194374

**Published:** 2021-09-24

**Authors:** Victoria Allgar, Barry Wright, Amelia Taylor, Ann Le Couter, Helen Phillips

**Affiliations:** 1Faculty of Health, Peninsula Medical School, University of Plymouth, Plymouth PL4 8AA, UK; 2Hull York Medical School, University of York, York YO10 5DD, UK; barry.wright@york.ac.uk; 3Leeds and York Partnership NHS Foundation Trust, Leeds LS15 8ZB, UK; amelia.taylor2@nhs.net (A.T.); helenphillips4@nhs.net (H.P.); 4Institute of Health and Society, Newcastle University, Newcastle NE1 7RU, UK; a.s.le-couter@newcastle.ac.uk

**Keywords:** autism spectrum disorder, deaf, child, young person, assessment, diagnosis, play based assessment, semi-structured interview

## Abstract

The aim was to investigate the agreement between the ADI-R Deaf adaptation and ADOS-2 Deaf adaptation overall diagnostic categorisation for autism (AUT) and a wider threshold to include autism spectrum (ASD) in a cohort of deaf children with and without ASD. We compared results of the instruments used on their own and when combined and propose standard criteria for the combined use of the ADI-R Deaf adaptation and ADOS-2 Deaf adaptation for use with deaf children. In total, 116 deaf children had a Gold standard NICE guideline assessment; 58 diagnosed with ASD and 58 without ASD, and for both groups a blinded informant based ADI-R Deaf adaptation and direct assessment using the ADOS-2 Deaf adaptation were separately completed. There was moderate agreement between the ADI-R Deaf adaptation and ADOS-2 Deaf adaptation for the wider threshold of ASD (Kappa, 0.433). To achieve the lowest number of false negatives, the most successful assessment tool approach is using the wider threshold of ASD with either ADI-R Deaf adaptation or ADOS-2-Deaf adaptation (95% sensitivity). This compares with 88% for the ADI-R Deaf adaptation alone and 74% for the ADOS-2-Deaf adaptation alone (wider threshold of ASD). To achieve a low number of false positives, the most successful assessment tool approach is a combination of ADI-R Deaf adaptation and ADOS-2- Deaf adaptation (using the narrow threshold of autism for both) (95% specificity). This compares with 83% for the ADI-R Deaf adaptation alone and 81% for the ADOS-2-Deaf adaptation (narrow threshold) alone. This combination is therefore recommended in specialist clinics for diagnostic assessment in deaf children.

## 1. Introduction

Autism spectrum disorders (ASD) occur in approximately 1.5% population and can occur with or without intellectual disability [1]. The current internationally agreed diagnostic criteria for ASD require evidence of differences/difficulties in two domains [2] namely social communication [3] and restricted and repetitive behaviours (RRB) [4]. A multi-disciplinary diagnostic assessment would include observations and information from multiple sources, such as parents/carers, and teachers. A number of measures have been developed to assist the systematic gathering of information such as using a semi-structured interview with an expert informant (e.g., a primary care-giver) to undertake a developmental history and through direct observation of play, activities and interaction with the child or young person themselves. Beyond parent surveys, there are no community prevalence studies of ASD in deaf children. Several studies have reported that deaf children can be difficult to assess for ASD because of confusion about the causation of behaviours and symptoms [5], lack of confidence and expertise in clinicians, and lack of adequate assessment tools [6]. In particular, deaf children without ASD but with language deprivation, where children have reduced language learning opportunities in early years, commonly present with social and emotional developmental delays [7]. This may include for example lack of awareness of or sensitivity towards communicative partners [8], delays in self-regulation [9], and difficulties in social relationships [10].

Two commonly used standardized assessment tools are a parent semi-structured interview, the autism diagnostic interview revised (ADI-R) [11] and a play and interaction based assessment the autism diagnostic observation schedule version 2 (ADOS-2) [12]. They have recently been adapted for use with deaf individuals [13,14]. Previous research in hearing populations has found that specificity improves when both instruments are combined [15,16]. This reduces the number of false positives identified, which can prevent unnecessary additional assessment and parent/carer anxiety. This study investigates the level of agreement between the newly validated ADI-R Deaf adaptation and ADOS-2 Deaf adaptation compared to the clinical diagnostic categorisation for autism or ASD, in a cohort of deaf children and explores whether or not sensitivity and/or specificity improves if these instruments are used in combination. Through this, we set out to provide helpful parameters in clinical practice for the combined use of the ADI-R Deaf adaptation and ADOS-2 Deaf adaptation in the assessment and diagnosis of ASD in deaf children.

## 2. Materials and Methods

### 2.1. Participants and Recruitment Methods

The data used in this research were drawn from a larger study, which aimed to validate assessment instruments for ASD in deaf children [13]. The study was reviewed and approved by Research and Development at Leeds and York Partnership NHS Foundation Trust (LYPFT) and an ethical approval was obtained from National Research Ethics Service (NRES) Committee Yorkshire & the Humber-South Yorkshire. REC reference: 15/YH/0093. The study recruited deaf children with and without a diagnosis of ASD between ages 2 and 18 who were assessed as moderate to profoundly deaf (bilateral; 40 dBHL or more). Children with intellectual disability or other health or mental health co-morbidities were included (Table 1) and there were no exclusions on preferred form of English language or sign language used (e.g., spoken English, Sign Supported English (SSE), British Sign Language (BSL), or a combination of these). Recruitment was across England through contact with all schools for the deaf, mainstream schools with specialist resource bases for deaf children and special educational needs schools, and a national register of Teachers of the Deaf. The study team also contacted the 10 National Deaf Child and Adolescent Mental Health Services (CAMHS) [17] and other CAMHS services across England, and asked them to circulate details of the study to potentially eligible children and their families. Organisations such as the National Autistic Society, National Deaf Children’s Society (NDCS), the national ASD UK, and the Database of Children Living with Autism Spectrum Disorder in the North East (Daslne) databases shared study information with their members. Parents/guardians and/or young people gave fully informed consent/assent to take part in the study. Deaf children without ASD were largely recruited through teachers of the deaf, deaf schools and deaf units, the NDCS and social media.

### 2.2. Diagnostic Procedure

National Institute for Health and Care Excellent (NICE) guideline standard clinical assessments based on World Health Organisation Research Diagnostic Criteria for ASD [18], were carried out by experienced senior multidisciplinary child mental health clinicians from the UK specialist national deaf child and adolescent mental health service (NDCAMHS) [17]. The clinicians met with parents to gather a comprehensive developmental history, observed and interacted with children at home or school, and viewed additional professional reports (e.g., teacher, speech and language therapy and educational psychology), including a parent completed social communication questionnaire (SCQ) [19]. The information was collated using a reporting matrix, which indicated whether the child’s profile would meet the criteria for a diagnosis of ASD based on the ICD-10 research diagnostic criteria ASD [18]. These assessments were conducted blind to the deaf adaptation research assessments (ADOS-2 Deaf adaptation and ADI-R Deaf adaptation) described below, which were completed by different clinicians on a different day.

### 2.3. Sample Size

The sample size for this research was pre-specified as 65 per group, based on estimating the difference in mean scores on the questionnaire scores between deaf children with ASD and deaf children without ASD to within ±0.34 standard deviations (95% confidence interval on each side of the estimate).

### 2.4. Measures

The Autism Diagnostic Observation Schedule-2 Deaf adaptation (ADOS-2 Deaf adaptation) is a play-based assessment formed around a series of play and interaction-based activities between a trained assessor and the child being assessed. A Delphi International Consensus Panel (DIEP) modified and validated it for use with deaf children and young people [14,20]. The finalised ADOS-2 Deaf adaptation was conducted by clinicians who were already trained in ADOS-2 and had top up training for the deaf adaptation, or by clinicians who completed a bespoke 5-day training course on the ADOS-2 Deaf adaptation. [14].

The Autism Diagnostic Interview-Revised Deaf adaptation (ADI-R Deaf adaptation) is a semi-structured interview administered by a trained interviewer with a parent or carer, modified and validated for use with deaf children and young people [20]. The interview focuses on functional domains such a language and communication, social interactions and restricted and repetitive behaviours. The ADI-R deaf adaptation interviews were conducted by experienced clinicians from the national deaf child and adolescent mental health service with ten centres across England. They all received additional specialist training on how to administer the ADI-R deaf adaptation [20] and the ADOS-2 deaf adaptation [14]. Further details are described in those publications.

The ADOS-2 Deaf adaptation uses diagnostic algorithms for each module based on the original instruments and the selection of module is determined by the language level of the participant. For the ADOS-2 Deaf adaptation, algorithm thresholds are reported in this study on the social reciprocity and communication domains and total scores in line with the original ADOS-2 guidelines. The justifications for the use of these thresholds are described more completely elsewhere [14]. The classifications are autism (AUT) using a narrow threshold, autism spectrum (ASD) using a wider threshold or non-spectrum (NS) and require participants to exceed thresholds in areas of social reciprocity, language and communication and repetitive behaviours, as well as have evidence of onset before 36 months, and are described more completely elsewhere [20]. The AUT thresholds are more stringent than the ASD thresholds. These follow the original parent instruments [11,12]. Non-spectrum means that there is limited or no evidence of ASD.

### 2.5. Analyses

The levels of agreement between the ADI-R Deaf adaptation and ADOS-2 Deaf adaptation were compared by calculating Kappa statistics [21] with respect to overall diagnostic categorisation of autism for each measure. The results for single and combined ADI-R Deaf adaptation and ADOS-2 Deaf adaptation algorithms were compared with the NICE guideline standard clinical assessment used in this context as the ‘gold standard’. Sensitivities and specificities for single and combined use of the ADI-R Deaf adaptation and ADOS Deaf adaptation algorithms were compared with NICE guideline standard clinical assessment diagnosis. A test is evaluated by the extent to which it identifies individuals with the disorder (high sensitivity) and excludes those without the disorder (high specificity) [22,23]. For the sensitivities and specificities, 95% confidence intervals are presented.

Each instrument identifies two main thresholds of combined symptom severity described as autism (AUT) (more severe) and autism spectrum (ASD) (less severe) and based on the original algorithms defined in previous research [11,12,15] used in deaf participants [14,20]. Seven combinations are therefore available for comparison using the ADI-R Deaf adaptation alone (2 threshold levels AUT and ASD), the ADOS-2 Deaf adaptation alone (AUT and ASD), or a combination of both together, ordered from the most stringent (requiring autism diagnoses (AUT) from both instruments) to the least stringent (requiring at least autism spectrum diagnosis (ASD) from ADOS-2 Deaf adaptation or ADI-R Deaf adaptation).

## 3. Results

Several of the initial target of 130 children and young people did not complete all three assessments because of practical constraints, illness, or availability. One hundred and sixteen children/young people and their primary care-giver completed the NICE guideline standard clinical assessment and both blinded assessments (each administered by different clinicians) using the ADI-R Deaf adaptation and ADOS-2 Deaf adaptation. There were 58 ASD children and 58 no ASD children based on the NICE guideline standard assessment. The ASD children were slightly older, more likely to be male and a higher proportion of Asian and mixed ethnic group (Table 2). Table 3 shows the classification based on the ADI-R and ADOS-2 deaf adaptations thresholds by ASD diagnosis based on NICE guidelines. There was moderate agreement between the ADI-R Deaf adaptation and ADOS-2 Deaf adaptation for the wider threshold of ASD (Kappa, 0.433).

Table 4 shows that the highest sensitivity (95%, 95% Confidence Interval (CI): 86%, 99%) was seen with either ADI-R Deaf adaptation or ADOS-2-Deaf adaptation wider threshold of autism spectrum, whereas lower sensitivities were seen when requiring the narrow threshold of autism for ADOS-2. This compares with 88% (95% CI: 77%, 95%) for the ADI-R Deaf adaptation alone and 74% (95% CI: (61%, 85%) for the ADOS-2-Deaf adaptation alone (wider threshold).

To achieve the highest specificity for a narrow threshold of AUT, the most successful assessment tool approach is a combination of ADI-R Deaf adaptation (narrow threshold) AND ADOS-2-Deaf adaptation (narrow threshold) (95%, 95% CI: 86%, 99%). This compares with 83% (95% CI: 71%, 91%) for the ADI-R Deaf adaptation alone and 81% (95% CI: 69%, 90%) for the ADOS-2-Deaf adaptation (narrow threshold) alone.

## 4. Discussion

Sensitivity and specificity are essential indicators of test accuracy and allow healthcare providers to determine the appropriateness of the diagnostic tool [23].

Both sensitivity and specificity are improved when instruments are used together compared to their use separately. The findings from this study show that the highest sensitivity (95%, 95% CI: 86%, 99%), the lowest false negatives, was seen when requiring at least autism spectrum (wider threshold) from ADOS-2 Deaf adaptation OR ADI-R Deaf Adaptation (wider threshold). High sensitivity leads to few false negative results where few actual cases are missed [23,24]). This combination may be helpful when identifying children in schools in order to plan interventions and monitor progress. To achieve the highest specificity, the lowest false positives, the most successful joint assessment tool approach was ADI-R Deaf adaptation (narrow threshold) AND ADOS-2-Deaf adaptation (narrow threshold). This combination of measures may be helpful in a diagnostic clinic to reduce the risk of an inappropriate ASD diagnosis [24]. In this way, the instruments can be used together for different purposes, depending on where in the care pathway they are used and whether it is preferable to have high sensitivity or specificity.

Previous research with hearing children supports this approach. A systematic literature review by Falkmer and colleagues [25] reviewed the accuracy, reliability, validity and utility of reported diagnostic tools and assessments. They identified 11 studies that assessed the ADI-R combined with the ADOS. The results show that the ADOS and ADI-R together have a correct classification rate for autism (narrow definition) of 0.88 in children under 3 years and 0.84 for children over 3 years.

Previous authors, Le Couteur et al. (2008) and Ventola et al. (2006) [26,27], have recommended the use of both instruments together as they provide information from different sources in a complementary way. Both studies explored the combined use of the measures in a sample of pre-school age children and highlighted the need for clinicians using multiple sources when diagnosing a child. Gray and colleagues (2007) [28] also explored the diagnostic validity of the ADI-R and ADOS in pre-school age children. This study reports that the ADI-R did not perform as well as the ADOS in this sample, but that both instruments provide important information in the clinical assessment of ASD and are best when used in conjunction with each other. Given hard-pressed clinical time in most centres, further clinical research could explore circumstances where different assessment tools are used in combination for different groups of children, exploring improved ways of allocating resources.

The benefits of using both instruments are also seen in older children. deBildt and colleagues [29] studied the interrelationship between the ADI-R and the ADOS-Generic (ADOS-G). They studied 184 children with learning disabilities aged 5–8 and over 8 years. They reported that a combined use of the ADI-R and ADOS-G identifies autism most appropriately, especially in younger children (age 5–8).

The kappa correlation between the ADI-R autism spectrum threshold and ADOS-2 autism spectrum threshold tends to vary between studies and can be relatively low (e.g., 0.54 in a study of UK preschool children [25] and −0.066 in a US study of toddlers [26] with our study finding a kappa of 0.433. However, both Risi and colleagues, 2006, and Kim and Lord 2012, [15,16] found that specificity improves when both instruments are combined together; this finding was confirmed for this sample of deaf children. This is important as it reduces the number of deaf children incorrectly identified with ASD. Accurate diagnosis as part of a diagnostic formulation of a child and family’s strengths and needs will in turn inform the care pathway and, importantly, education provision, so that, for example, a deaf child without ASD (but with language or socio-emotional developmental delays) is given appropriate education provision and not placed in an autism education unit.

There are no dedicated autism education units for deaf children in the UK or indeed in most international countries. Difficult decisions are made about where children are educated with the main choices being a deaf unit within a mainstream school, a deaf residential school or an autism unit in a special school or a mainstream school. Many deaf children with early life language deprivation may have social communication problems possibly mediated by theory of mind delays [7]. This is a feature of ASD [30] and so deaf children with these difficulties may be confused with children who have ASD [5]. These children do not have ASD, but may well be delayed in empathy skills by 2–3 years. They need exposure to other children for social interaction and social experiences to support social and emotional development. Given large challenges for education systems and schools meeting the needs of deaf children with ASD [31], and the risk of underestimating any children diagnosed with ASD in school settings [32], there is a large risk for placing a deaf child without ASD in an ASD setting, for example, related to opportunities to develop social skills. Hence, accurate diagnosis is important.

Strengths of this study include that this is the first study comparing a large number of deaf children who had received both these measures. The sample being partly drawn from deaf clinical services in ten centres across England could also be said to be representative of deaf children in the UK, although did not include many children aged 3 and under. Another strength was the comparison with a separate and blinded NICE guideline standard assessment. It was also a strength that a large number of children with a variety of co-morbidities were included demonstrating an open pragmatic recruitment strategy and reflecting practice in the real world.

Limitations of this study include the fact that it was carried out in the UK, mainly with participants using English, Sign Supported English, and BSL or combinations of these, and results may be different in other countries with differing cultures or languages. The numbers were slightly below our targets in each group because of the timescales and complexities organizing multiple assessments in the community. We boosted these by increasing the recruitment period and improving our marketing reach (e.g., by using social media). Further validation research in a larger sample would be helpful.

## 5. Conclusions

In summary, this study shows that the benefits found when combining the ADI-R and the ADOS-2 in assessments of children in the general population apply in deaf children. They allow us to combine important information from parents about their child’s development with information from direct play and interaction with the child. These tools for ASD assessment can be available for use with deaf participants, and given the large struggles experienced by parents seeking assessment [33] and by assessing clinicians [6], this is a step forward. Whilst these assessments may be used in specialist centres, with deaf and hearing clinicians working together [17], this research has opened up new assessment possibilities. With training and the availability of these deaf adaptations, assessments could be undertaken in community child health/mental health centres supported by specialist teams, such as National Deaf CAMHS. Further work could be done to explore a model of service provision that is realistic, pragmatic, and effective in the context of current services, and might be helpful in other countries and cultures.

## Figures and Tables

**Table 1 jcm-10-04374-t001:** Co-morbidities of participants in combined use of ADOS-2 Deaf adaptation and ADI-R Deaf adaptation validation study.

Clinical Diagnoses *	Deaf Children with ASD (*n* = 58) *n* (%)	Deaf Children without ASD (*n* = 58) *n* (%)
**Genetic**		
Alport syndrome	1 (1.7)	1 (1.7)
Branchio-oto-renal syndrome	0 (0)	2 (3.4)
Chromosome 1q21.1 duplication syndrome	1 (1.7)	0 (0)
Chromosome 15 deletion	0 (0)	1 (1.7)
Connexin 26	3 (5.2)	4 (6.9)
Mondini dysplasia	1 (1.7)	0 (0)
Pendred syndrome	2 (3.4)	2 (3.4)
X linked Stapes Gusher Syndrome	0 (0)	1 (1.7)
Waardenburg	2 (3.4)	4 (6.9)
XXY (Klinefelter syndrome)	1 (1.7)	0 (0)
**Developmental**		
Dyspraxia/motor co-ordination disorder	5 (8.6)	4 (6.9)
Learning Disability	14 (24.1)	7 (12.1)
Language Delay	18 (31)	13 (22.4)
Von Hippel-Landau disease	1 (1.7)	0 (0)
**Neurological/physical**		
Asthma	6 (10.3)	2 (3.4)
Aural atresia	1 (1.7)	0 (0)
Disorder of Vestibular Function	0 (0)	2 (3.4)
Cerebral palsy	0 (0)	2 (3.4)
Past Cytomegalovirus infection	4 (6.9)	1 (1.7)
Epilepsy	1 (1.7)	3 (5.2)
Hypermobility	1 (1.7)	0 (0)
Microcephaly	0 (0)	2 (3.4)
Multiple physical problems	5 (8.6)	1 (1.7)
Sensory processing disorder	6 (10.3)	4 (6.9)
Visual problems (more than acuity)	1 (1.7)	2 (3.4)
**Mental health**		
Attention Deficit Hyperactivity Disorder	5 (8.6)	4 (6.9)
Conduct/Behaviour	1 (1.7)	3 (5.2)
Emotional (serious anxiety disorders)	6 (10.3)	6 (10.3)

* Parent reported clinical diagnoses.

**Table 2 jcm-10-04374-t002:** Demographic characteristics by clinical interview diagnosis.

	ASD *n* = 58	No ASD *n* = 58
**Age**	9.6 (4.5)	7.5 (3.5)
**Gender**		
Male	50 (86%)	41 (71%)
Female	8 (14%)	17 (39%)
**Ethnicity**		
White	43 (74%)	53 (91%)
Black	1 (2%)	2 (3%)
Asian	7 (12%)	2 (3%)
Mixed	6 (10%)	1 (2%)
Other	1 (2%)	0 (0%)

**Table 3 jcm-10-04374-t003:** Classification based on ADI-R Deaf adaptation and ADOS-2 Deaf adaptation thresholds by ASD diagnosis based on NICE guideline standard clinical assessment.

	ASD *n* = 58	No ASD *n* = 58
**ADOS-2 Deaf adaptation**		
Non spectrum	15 (26%)	42 (72%)
Autism spectrum	5 (9%)	5 (9%)
Autism	38 (65%)	11 (19%)
**ADI-R Deaf adaptation**		
Non spectrum	7 (12%)	48 (83%)
Autism	51 (88%)	10 (17%)

**Table 4 jcm-10-04374-t004:** Sensitivity and specificity of ADI-R Deaf adaptation and ADOS-ASD thresholds by the NICE guideline standard clinical assessment for the comparator assessment.

	No of True Positives	No of False Negatives	No of True Negatives	No of False Positives	Sensitivity (95% CI)	Specificity (95% CI)
ADI-R Deaf adaptation (narrow threshold)	51	7	48	10	88% (77%, 95%)	83% (71%, 91%)
ADOS-2 Deaf adaptation (narrow threshold)	38	20	47	11	66% (52%, 78%)	81% (69%, 90%)
ADOS-2 Deaf adaptation (wider threshold)	43	15	42	16	74% (61%, 85%)	72% (59%, 83%)
ADI-R Deaf adaptation and ADOS-2 Deaf adaptation (narrow threshold)	35	23	55	3	60% (47%, 73%)	95% (86%, 99%)
ADI-R Deaf adaptation and ADOS 2 Deaf adaptation (wider threshold)	39	19	53	5	67% (81%, 97%)	91% (81%, 97%)
ADI-R Deaf adaptation or ADOS-2 Deaf adaptation (narrow threshold)	54	4	40	18	93% (83%, 98%)	69% (55%. 80%)
ADI-R Deaf adaptation or ADOS-2 Deaf adaptation (wider threshold)	55	3	37	21	95% (86%, 99%)	64% (50%, 76%)

NB; ‘or’ refers to thresholds being met for one or the other instrument; ‘and’ refers to thresholds being met for both instruments.

## Data Availability

Ethics approval allows for data to be kept for 10 years. Access to limited anonymised data can be requested from the Trial Management team via the corresponding author subject to meeting agreed ethical approvals.

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
