# Peer review of "Diagnosing Autism Spectrum Disorders in Deaf Children Using Two Standardised Assessment Instruments: The ADIR-Deaf Adaptation and the ADOS-2 Deaf Adaptation"

_jcm, 2021, doi:10.3390/jcm10194374_

Round 1
Reviewer 1 Report
The authors propose standard criteria for the combined use of the ADI-R Deaf adaptation and ADOS-2 Deaf adaptation for use with deaf children after investigating the agreement between both instruments in a cohort of deaf children with and without ASD. The authors perform a well designed and written study with a large sample difficult to collect that shows the efforts and collaboration of multiple agents. The aim is an important one and provides useful guidelines for clinical and educative decisions. The limitations in the study have been already acknowledged by the authors.
My suggestion is to include in Table 3 the Kappa values, it will serve also to compare with previous results mentioned in the discussion section.
Reviewer 2 Report
Dear Authors,
your study is very important with regard to the identification of deaf children with autism spectrum disorder. However, I found many parts of the manuscript difficult to understand. Please provide more details on study procedure, recruitment, inclusion of participants, etc. in the methods section. Some terms need to be more clearly explained when they first occur and should then be consistently used throughout the manuscript so that the reader can follow the manuscript (e.g., narrow/wide threshold, AUT, ASD, NS,...). The interpretation of the results seems to be incomplete.
Please find here my detailed comments:
Title/throughout manuscript:
Please justify why you use "autism spectrum disorders" instead of "autism spectrum disorder".
Introduction:
It would be interesting for the reader to know the prevalence of ASD in deaf children - please add some details regarding this issue - is it similar to the general population?
Line 38: There seems to be a formatting error after the reference.
Materials and Methods:
Please explain how you come to the final number of participants. This is very unclear from the methods section. At first you speak of 204 deaf children, then you say 65 per group - please also define which groups there are - and in the table you have 58 with ASD and 58 without ASD. How did you recruit the children without ASD? Do they have other medical issues apart from being deaf and how did you determine this? Please add a table or similar to get an overview on the number of children with which co-morbidities. It is not clear whether these co-morbidities occurred in both groups.
Please define the term "NICE" in NICE guideline standard clinical assessments.
Line 109: Is there a formatting error in pre-specified?
Please add more details on the study procedures: e.g., were the assessments conducted on the same day and by the same/how many clinicians?
Line 135: Please check grammar.
Line 133/134: It is not clear what is meant here - please rephrase the sentence for clarification.
Line 138/139: Please explain the classifications autism, Autism Spectrum, non-spectrum is detail. What is the difference between these categories? This is not clear to the reader, but very important to understand the manuscript.
Line 141: What exactly do you mean with "autism" here? Your classification "AUT"?
Line 150: Do you mean the classification AUT here or AUT+ASD+NS?
Line 160/161: Please explain in more detail what you mean with more/less severe scores.
Results:
The first chapter seems to be rather part of the methods section.
Please refer and explain Table 2 in the text.
Line 223: please define CI.
Line 225/226: Please be more concrete here: Do you mean either ADI-R Deaf or ADOS-2 Deaf alone with narrow threshold?
Line 226ff. and Line 232ff.: It is not clear for me why you reported the selected results here, but not others from your table.
Table 3: Please explain abbreviations in the table caption. Also explain what the numbers in parenthesis mean.
Discussion: The discussion of your findings is very short. In paragraph 2 you mention that highest sensitivity was achieved via ADOS-2 Deaf adaptation OR ADI-R Deaf Adaptation and highest specificity was achieved via a combination of ADI-R Deaf (narrow) and ADOS-2 Deaf (narrow). Is it right that a combination of the assessments results in low sensitivity of 60 or 67%? If not, please clarify in the text. Please add a thorough discussion about what these findings add to the current state-of-the-art and the implications they may have for the clinical practice. Please also interpret your findings in light of the co-morbidities of the participating children and also in the light of the mentioned classifications AUT, ASD, NS.
Line 270: Please rephrase "looked at".
Line 276/277: Please check citation style.
Line 306ff. This needs to be explained already in the methods section - otherwise the reader is confused about the different numbers.
Conclusion: What about your finding that ADI-R Deaf adaptation and ADOS-2 Deaf adaptation alone have higher sensitivity than the combination? Please discuss this and explain why you think your results suggest the combination of the assessment tools.
Author Response
|
Reviewer 2: Please find here my detailed comments: Title/throughout manuscript: Please justify why you use "autism spectrum disorders" instead of "autism spectrum disorder".
Introduction: It would be interesting for the reader to know the prevalence of ASD in deaf children - please add some details regarding this issue - is it similar to the general population? Line 38: There seems to be a formatting error after the reference.
Materials and Methods: Please explain how you come to the final number of participants. This is very unclear from the methods section. At first you speak of 204 deaf children, then you say 65 per group - please also define which groups there are - and in the table you have 58 with ASD and 58 without ASD. How did you recruit the children without ASD? Do they have other medical issues apart from being deaf and how did you determine this? Please add a table or similar to get an overview on the number of children with which co-morbidities. It is not clear whether these co-morbidities occurred in both groups.
Please define the term "NICE" in NICE guideline standard clinical assessments. Line 109: Is there a formatting error in pre-specified? Please add more details on the study procedures: e.g., were the assessments conducted on the same day and by the same/how many clinicians? Line 135: Please check grammar. Line 133/134: It is not clear what is meant here - please rephrase the sentence for clarification. Line 138/139: Please explain the classifications autism, Autism Spectrum, non-spectrum is detail. What is the difference between these categories? This is not clear to the reader, but very important to understand the manuscript. Line 141: What exactly do you mean with "autism" here? Your classification "AUT"?
Line 150: Do you mean the classification AUT here or AUT+ASD+NS?
Line 160/161: Please explain in more detail what you mean with more/less severe scores.
Results: The first chapter seems to be rather part of the methods section.
Please refer and explain Table 2 in the text. Line 223: please define CI. Line 225/226: Please be more concrete here: Do you mean either ADI-R Deaf or ADOS-2 Deaf alone with narrow threshold? Line 226ff. and Line 232ff.: It is not clear for me why you reported the selected results here, but not others from your table. Table 3: Please explain abbreviations in the table caption. Also explain what the numbers in parenthesis mean.
Discussion: The discussion of your findings is very short. In paragraph 2 you mention that highest sensitivity was achieved via ADOS-2 Deaf adaptation OR ADI-R Deaf Adaptation and highest specificity was achieved via a combination of ADI-R Deaf (narrow) and ADOS-2 Deaf (narrow). Is it right that a combination of the assessments results in low sensitivity of 60 or 67%? If not, please clarify in the text. Please add a thorough discussion about what these findings add to the current state-of-the-art and the implications they may have for the clinical practice. Please also interpret your findings in light of the co-morbidities of the participating children and also in the light of the mentioned classifications AUT, ASD, NS.
Line 270: Please rephrase "looked at". Line 276/277: Please check citation style. Line 306ff. This needs to be explained already in the methods section - otherwise the reader is confused about the different numbers.
Conclusion: What about your finding that ADI-R Deaf adaptation and ADOS-2 Deaf adaptation alone have higher sensitivity than the combination? Please discuss this and explain why you think your results suggest the combination of the assessment tools.
|
Response:
We have used the convention used in many previous published pieces of research including the following referenced papers in our article: 1,6,15, 16,19,25 and 26.
There are no prevalence studies because until now there have been no validated assessment tools. We have added this in the first paragraph.
We have amended this error.
The peer reviewer is correct to say that we identified 130 target children (65 x2) and that only 116 completed all three assessments necessary for analysis. We have explained this at the beginning of the results section.
We have added in a sentence about recruitment of deaf participants without ASD : ‘Deaf children without ASD were largely recruited through teachers of the deaf, deaf schools and deaf units, the NDCS and social media.’
We have now added a comorbidity table. We agree this adds to the paper.
This has been added
This has been amended
We have added that these were completed by different clinicians on a different day, blind to each other.
Amended
We have changed this sentence so that it is clearer.
We have explained these differences in the text.
We have now explained in the text that AUT (autism) involves a child who has problems in all domains of diagnosis including social reciprocity, language and communication and repetitive and stereotyped behaviours.
Table 4 outlines the various different ways the data has been analysed to compare when AUT and ASD are or are not combined. We believe that this lays it out clearly.
We have added to this sentence to explain where these two thresholds came from. They exist in the published ADI-R and ADOS-2 algorithms. We have explained this.
This first section describes the numbers of children and some of their demographics and so we believe it fits in the results section.
We have added this in the text.
We have defined this.
Done
We picked out the key findings from Table 4, but did not want to replicate all the Table 4 results.
We have added these in.
The best sensitivity of the combined instruments was 95% which is very good. We have tried to make this clear. 60% is only found if the two instruments are used in conjunction in one particular way. We presented the range of ways the instruments could be combined for completeness of data but we would not suggest it would be used with a sensitivity of 60%.
We have added some sentences describing the clinical uses of the instruments used in different ways in combination.
We have added a section suggesting the large number of co-morbidities included, showing real world practice.
We have changed this to ‘studied’
This has been amended
We are not sure where the peer reviewer has gleaned this information. We could not find numbers at this location.
The sensitivity is highest in combination: 95% in combination compared to 88% ADI-R alone.
The specificity is highest in combination: 95% compared to 83% of ADI-R alone.
We have added this into the discussion (line 253). To make this clear we have also added a footnote to table 4 : NB. ‘or’ refers to thresholds being met for one or other instrument. ‘and’ refers to thresholds being met for both instruments
|

Round 2
Reviewer 2 Report
Thanks for your reply! The manuscript is much clearer now. I appreciate that you have added a table with the co-morbidities.